# Advancing Neural Network Performance through Emergence-Promoting Initialization Scheme

## Abstract

Emergence in machine learning refers to the spontaneous appearance of complex behaviors or capabilities that aris from the scale and structure of training data and model architectures, despite not being expicitly programmed. We introduce a novel yet straightforward neural network initialization scheme that aims at achieving greater potential for emergence. Measuring emergence as a king of structural nonlinearity, our method adjusts the layer-wise weight scaling factors to achieve higher emergence values. This enhancement is easy to implement, requiring no additional optimization steps for initialization compared to GradInit. We evaluate our approach across various architectures, including MLP and convolutional architectures for image recognition, and transformers for machine translation. We demonstrate substantial improvements in both model accuracy and training speed, with and without batch normalization. The simplicity, theoretical innovation, and demonstrable empirical advantages of our method make it a potent enhancement to neural network initialization practices. These results suggest a promising direction for leveraging emergence to improve neural network training methodologies.

## 1 Introduction

Emergence, in general, refers to the phenomenon where complex behaviors and properties arise from the interactions of simpler elements within a system. In machine learning, emergence has been studied as the nonlinear increase in system performance as the system's size increases, exemplified by the emergent abilities of large language models. These emergent behaviors are crucial for enabling neural networks to perform complex tasks such as image recognition, natural language processing, and strategic game playing (Brown, 2020; Kaplan et al., 2020; Radford et al., 2019).

Although the concept of emergence has been observed in various fields and disciplines—such as phase transitions in physics and emergent structures and functions in biological networks—a unifying trait of these emergent phenomena is their association with nonlinearity. Generalizing the notion of a nonlinear function in calculus, this nonlinearity implies the disproportionate increase in system behavior when moving from the partial to the overall structure of the system.

A natural question regarding emergence is: what kind of system has a stronger potential for emergence? It is generally appealing to link the emergent function with the structure. To address this question, (Li et al., 2023) developed a measure of emergence based on network structure. This measure, which quantifies how much emergence a system can sustain, suggests a design principle for neural networks, enabling us to tune the network structure to maximize emergence.

Based on this measure, we propose a neural network initialization scheme that encourages emergence. The initialization of network parameters significantly impacts the training stability and performance of deep neural networks. Initializations that prevent gradient explosion or vanishing during backpropagation played a key role in the early successes of feed-forward networks (Glorot & Bengio, 2010; He et al., 2015). However, it remains theoretically challenging to link a network's initialization with its training dynamics, especially for structure- and dataset-agnostic initialization schemes (Glorot & Bengio, 2010; He et al., 2015; Saxe et al., 2013; Mishkin & Matas, 2015; Zhu et al., 2021; Gilmer et al., 2021).

Our motivation differs from existing literature, which emphasizes network stability. By initializing networks with a stronger potential for emergence, we increase the likelihood of exhibiting emergent behaviors and patterns during training. This nonlinearity-based emergence suggests that the network structure and functionality are more susceptible to change, intuitively leading to larger training gradients.

We show that network architectures with stronger emergence, based on our measure, exhibit patterns of increasing activation, resembling natural emergent structures like dominos, where initial perturbations can lead to significant global changes, aligning with the general notion of emergence.

In this paper, we introduce a new initialization scheme for neural networks that leverages the concept of emergence. Our method adjusts layer-wise variance parameters to achieve higher emergence values compared to traditional methods like Xavier and Kaiming initialization (Glorot & Bengio, 2010; He et al., 2015). This approach is particularly appealing because it is straightforward to implement, requiring only minor modifications to existing initialization techniques without necessitating additional optimization or training steps.

Our initialization scheme is grounded in the idea that by enhancing the emergent properties of neural networks from the beginning, we can facilitate better feature differentiation and integration. This, in turn, can lead to improved performance across various tasks and architectures. We evaluate our method on both convolutional neural networks (CNNs) for image recognition and transformer architectures for machine translation, demonstrating significant improvements in model accuracy and convergence speed.

The simplicity and effectiveness of our approach make it a compelling addition to the toolkit of neural network initialization methods. By focusing on enhancing emergent properties, our scheme offers a new perspective on how initialization can impact the learning dynamics and ultimate performance of neural networks. This paper contributes to the growing body of research that seeks to understand and harness the power of emergence in machine learning, paving the way for more robust and capable models .

## 2 RELATED WORK

The initialization of neural networks has been a critical area of research, influencing the stability and speed of training, as well as the ultimate performance of the models. Traditional initialization schemes, such as Xavier (Glorot & Bengio, 2010) and Kaiming (He et al., 2015), have laid the foundation for effectively training deep networks by mitigating issues related to vanishing and exploding gradients. Xavier initialization aims to keep the scale of the gradients approximately the same in all layers, while Kaiming initialization, specifically designed for ReLU activations, helps to maintain the variance of activations throughout the layers. Both methods have proven to be fundamental in training deep networks but do not explicitly account for emergent properties within the networks.

Recent research has explored more sophisticated initialization strategies that leverage the structural and statistical properties of neural networks. For instance, (Saxe et al., 2013) studied the dynamics of signal propagation in deep networks, highlighting the importance of properly scaling the initial weights to ensure efficient training. Additionally, (Mishkin & Matas, 2015) proposed a layer-sequential unit-variance (LSUV) initialization that iteratively adjusts the weights to achieve unit variance across all layers, further improving convergence.

The concept of emergence, where complex behaviors arise from simple interactions within a system, has also been examined in the context of neural networks. Emergent properties have been shown to play a crucial role in the development of robust and adaptive models. Research by (Olah et al., 2020) illustrated how higher-level features and behaviors emerge in deep networks as a result of training on large datasets. This phenomenon underscores the potential for leveraging emergent properties to enhance network performance.

Despite these advancements, several challenges and limitations persist. Traditional initialization methods, while effective at preventing gradient-related issues, do not account for the complex emergent properties that can significantly influence network performance. More sophisticated methods, such as LSUV, improve convergence but may require iterative adjustments that complicate the initialization process.

The concept of emergence itself, although promising, is not yet fully understood or integrated into standard practices for network initialization. While studies like those by (Adam, 2017; Li et al., 2023) have made significant strides, there is still a need for practical methods that can harness emergent properties effectively from the outset of training.

Furthermore, the role of initialization in specific architectures such as transformers remains an area of active research. (Vaswani, 2017) introduced the transformer model, which has become a cornerstone in natural language processing due to its ability to capture long-range dependencies through self-attention mechanisms. However, recent research continues to refine transformer architectures, with improvements in initialization playing a crucial role in achieving state-of-the-art performance [15] (Liu et al., 2020).

## 3 METHOD

Emergence fundamentally arises from the observation of a system from a higher scale. We build our definition of emergence on the notion of nonlinearity as the information passed to higher scales. Two key conceptual components are necessary to qualitatively describe emergent effects within the framework proposed by (Adam, 2017). The first is a notion of interaction or local computation among the components of a system. For example, the communication and propagation of information among nodes or subnetworks in the neural network. The second is the notion of interactional effects, which equips each system with an observable, for example, attaches network with its performance or abilities. These kinds of interactional effects are almost always associated with partial observations, or a simplification and integration of lower — more foundational or granular —levels or scales in the system that result in a 'loss of information' or pattern/ feature formation at a higher level.

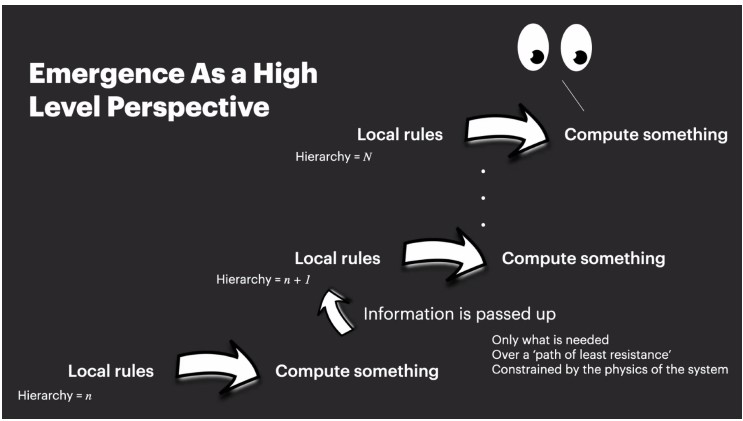

Figure 1: An illustration of emergence in the hierarchical system.

With these two ingredients, we can define emergence as a partial observation of interacting and interconnected components within a system that cannot be explained by known interactions that produce or result in partial observations of the components. This notion agrees with the intuitive understanding of emergence that some properties of the interconnected components cannot be decomposed or reduced to combinations of known properties of the constituent components, i.e. that the whole is more than the sum of its parts. This notion of emergence is the foundation on which our work in this paper, building on the framework first proposed in (Adam, 2017), develops a mathematical definition and computational measure of emergence.

To formalize these ideas, we begin by representing the interactions between components as an operation $\vee$, where $s_1 \vee s_2$ represents a new interconnected system of subsystems $s_1$ and $s_2$. Interactional effects are described by the mapping $\Phi$ that sends a system to its partial observation or interactional effect at a higher scale, in some cases corresponding to a coarse graining scheme (Rosas et al., 2024). Emergent effects are sustained whenever the observation of the combined system cannot be explained by the observation of the separate components. Mathematically,

**Definition** A system sustains emergent effects when the following inequality is satisfied:

$$\Phi(s_1 \vee s_2) \neq \Phi(s_1) \vee \Phi(s_2), \tag{1}$$

for some constituent subsystems $s_1$ and $s_2$.

This definition essentially captures emergence as a kind of "structural nonlinearity". Let's consider the simple case when $\Phi$ is simply a smooth function $f : \mathbb{R} \to \mathbb{R}$ and the interaction $\vee$ is simply taking the average, $s_1 \vee s_2 = (s_1 + s_2)/2$. Then we realize that the extent to each $\Phi(s_1 \vee s_2)$ differs from $\Phi(s_1) \vee \Phi(s_2)$ is just $\left| f\left(\frac{s_1+s_2}{2}\right) - \frac{f(s_1)+f(s_2)}{2} \right|$, which is related to how nonlinear the function $f$ is, and can be studied by the derivatives of $f$, in particular, the second order derivative, since $\left| f\left(\frac{s_1+s_2}{2}\right) - \frac{f(s_1)+f(s_2)}{2} \right|$ can be approximated by $\frac{|s_2-s_1|^2}{4}|f''(\xi)|$ for some $\xi$ between $s_1$ and $s_2$. Now when $\Phi$ is a functor [26], which captures the cross-scale information flow in real world systems, we want an analogue to derivatives to apply this idea, and this naturally leads to the concept of a derived functor in homological algebra[26]. We can also see that Definition captures the structural nonlinearity in emergence, the nonlinearity of system's behavior and functionality as the system's structure changes, or as we go from components to parts of the system to the whole system. This is a general mathematical definition of emergence, as first given in (Adam, 2017) [1]. Note that when studying the emergence of a specific system, the interaction $\vee$ and interactional effect $\Phi$ need to be chosen carefully.

**Examples of emergence in machine learning** Emergence or generativity has been a rising concept in machine learning, for example, (Wei et al., 2022a;b; Du et al., 2024). Emergent abilities of large language models, for example, (Wei et al., 2022a), commonly conceived as the new properties/ abilities of the larger models that do not exist in smaller models. If we consider $s_1$ and $s_2$ as two smaller models, $s_1 \vee s_2$ as combining two smaller models into a larger model by, for example, techniques in ensemble learning (Mohammed & Kora, 2023), and $\Phi$ as the mapping that reflects the properties/ abilities of the model, that is, $\Phi(s)$ is the ability acquired by the model $s$. Then $\Phi(s_1 \vee s_2)$ is the properties/ abilities of the combined model and $\Phi(s_1) \vee \Phi(s_2)$ can be interpreted as a summation of the properties/ abilities of each small model. Then the difference between $\Phi(s_1 \vee s_2)$ and $\Phi(s_1) \vee \Phi(s_2)$ can reflect the emergent properties/ abilities that result in the nonlinear increase of performance, related to the performance in (Wei et al., 2022a). The difference can also be related to generalizability, where $s_1$ and $s_2$ are two data sets, when the model trains on two data sets, it is usually different from training the model on separate datasets.

Based the mathematical theory of emergence in (Adam, 2017), the structural difference in (1) between $\Phi(s_1 \vee s_2)$ and $\Phi(s_1) \vee \Phi(s_2)$, can be evaluated through computing the mathematical structure of derived functor $R^1\Phi$, see (Adam, 2017; Rotman & Rotman, 2009). (Li et al., 2023) gives the following result that computes $R^1\Phi$, generalization of derivative, where the input is the mathematical structure of quiver representation.

**Theorem** (Proposition 5.3 in (Li et al., 2023)) Given the functor $\Phi$ which preserves partial structure in a quiver representation $W$ by deleting a set of edges $E$, the derived functor of $\Phi$ is

$$R^1\Phi(W) = \bigoplus_{a \in E} \Phi(W(ta) \otimes P_{ha}) \tag{2}$$

where $ta$ is the tail of edge $a$ (the starting node), $ha$ is the head of edge $a$ (the ending node), $W(ta)$ is the vector space associated to node $ta$, $P_{ha}$ is the vector space spanned by all paths originating from node $ha$.

*Proof.* The proof of this theorem is given in the appendix. □

This theorem computes $R^1\Phi$, which evaluates the difference between $\Phi(s_1 \vee s_2)$ and $\Phi(s_1) \vee \Phi(s_2)$, thus encodes the potential of a system for emergence. Taking advantage of this theorem, we can take the dimension of $R^1\Phi(W)$ as a numerical approximation of the potential for emergence of $W$ when

---

[1]In (Adam, 2017) the term "generativity" is used instead of emergence. These two terms are often considered interchangeable.

the network interact with other networks. Given a network $G$, and a sub-network $H$ which represents its effect or observation under $\Phi$, where their relation are shown as follows:

$$G \xrightarrow{\text{Cross-scale mapping } \Phi} H$$

then we have the following measure of emergence for networks:

$$\text{Emergence}(G, H) = \sum_{x \in G \backslash H} \#\text{paths in } H \text{ from } N_H(x) \text{ to } H, \tag{3}$$

where $H$ represents the part of network structure being preserved by $\Phi$, the partial observation. $N_H(x)$ is the set of downstream neighbors of $x$ in $H$.

Emergence in this context is inherently multiscale. It involves interactions across different scales of the network, where $G$ represents one scale and $H$ represents a higher scale. Emergence appears only when viewed from this multiscale perspective, as it captures the complexity arising from the network's hierarchical structure. In our graph-theoretical framework, the emergence value $E$ of a neural network is defined based on the number of paths from nodes at scale $G$ to nodes at scale $H$. This definition captures the essence of multiscale interactions within the network. The more paths that exist between these scales, the greater the degree of emergence.

- $G$ is the set of nodes at the lower scale,

- $H$ is the set of nodes at the higher scale.

Our measure captures the emergent behavior by accounting for the connections and interactions between different scales within the network. The higher the value of $E$, the more interconnected the network is across scales, thereby increasing the likelihood of complex behaviors and traits emerging from the network. Intuitively, a system with a higher value of $E$ has more extensive and interconnected pathways through which information can propagate across different scales. This interconnectedness facilitates the development of intricate patterns and features within the network, enabling it to capture and represent more complex relationships in the data. As a result, networks with higher emergence values are better equipped to learn and generalize from diverse and intricate datasets, leading to improved performance across various machine learning tasks.

Our approach leverages this definition to modify the initialization process of neural networks, aiming to enhance their emergent properties from the outset. By doing so, we achieved significant improvements in network performance, as demonstrated in our experimental results.

To understand emergence in the context of machine learning, when a model has stronger emergence traits, this means that the model is easy to learn any or certain downstream tasks. From a loss function perspective, a model with a stronger emergence should be closer to the global minimum, or the learning should be fast. We will show in our numerical experiments, that schemes with stronger emergence will indeed have faster learning in the initial epochs.

In machine learning setting, one modeling approach is to consider $\Phi$ as the training process, since emergence here evaluates the potential/ ability for emergent traits when we observe system $G$ from a higher level $H$, here we want $G$ to represent the model itself, and $H$ to be some certain features of the model. In the paper, we adopt the setting that $H$ is the nodes in $G$ that are still active in the training process, where the criteria for active nodes is the set of nodes whose average activation on all input data is greater than a threshold. This sorted out the nodes that are not actively participating in the computational process/ representing features. The set of active nodes thus in a sense represent the learning task, thus we can tie emergence with the performance of the network in a learning process. This fits in our framework of emergence, where part of the system is being neglected after the learning process, thus the learning process represents the $\Phi$ where partial observation is carried out, and the properties of $H$ represents the emergent abilities of the network.

For a feedforward network, with $N$ layers, and $n_i$ nodes for each layer, and $a_i$ the number of active nodes for each layer, emergence is computed as follows:

$$
\begin{aligned}
E &= \sum_{i=1}^{N-1} \sum_{j>i}^{N} \#\text{paths from inactive nodes in layer } i \text{to active nods in layer } j \\
&= \sum_{i=1}^{N-1} \sum_{j>i}^{N} (n_i - a_i) a_{i+1} \cdots a_{j-1} a_j \\
&= \sum_{i=1}^{N-1} \sum_{j>i}^{N} (n_i - a_i) a_j \prod_{k=i+1}^{j-1} a_k
\end{aligned}
\tag{4}
$$

when the network architecture is fixed, which means, when $L$ and $n_i, i = 1, \ldots, L$ are fixed, emergence is only a function of the number of active nodes in each layer,

$$
E := E(a_1, \ldots, a_N).
\tag{5}
$$

And the number of active nodes at initialization, is impacted by the weights. With a criteria for active nodes, for example, those nodes whose activation is greater than a threshold as adopted in this paper, we can thus establish initialization scheme that has stronger emergence.

**Lemma:** Emergence function $E(a_1, \ldots, a_N)$ increases when $a_1 \ldots a_i$ gets smaller and $a_{i+1} \ldots a_N$ gets larger, where $i$ is the largest integer such that

$$
-n_{i-1} + n_{i+1} + n_{i+1}n_{i+2} + n_{i+1}n_{i+2}n_{i+3} + \cdots + n_{i+1}n_{i+2} \cdots n_{N-1}n_N > 0.
\tag{6}
$$

Proof: Consider the case where all the layers before $i_{th}$ layer are fully inactive and all the layers after $i_t h$ layer are fully active, in other words, $a_k = 0$ for $k < i$ and $a_k = n_k$ for $k > i$. Then when $a_i$ decrease by 1, the number of paths from previous layers to layer $i$ will decrease by $n_{i-1}$, the number of paths from layer $i$ to latter layers will increase by $n_{i+1} + n_{i+1}n_{i+2} + n_{i+1}n_{i+2}n_{i+3} + \cdots + n_{i+1}n_{i+2} \cdots n_{N-1}n_N$. So the net increase of paths will be

$$
\Delta = -n_{i-1} + n_{i+1} + n_{i+1}n_{i+2} + n_{i+1}n_{i+2}n_{i+3} + \cdots + n_{i+1}n_{i+2} \cdots n_{N-1}n_N.
\tag{7}
$$

In a wide range of image recognition tasks, we propose to choose $i \approx N/2$ as it works for a wide range of neural network architecture blocks.

Let us consider now why the network with the configuration above has stronger emergence: for example, when the network is doing an image recognition task, the nodes in the later half of the layers are making important decisions on which category the image belongs to, so it needs to be more active/ subject to more sensitive weight changes. However, for the nodes in the initial half of the layers, they are subject to greater weight changes, and they could be turned off to represent some global features of the image. Hence they could be more inactive/ subject to more drastic weight changes.

This idea also agrees with the fine-tuning idea: typically, the initial layers (closer to the input) have smaller learning rates, while the later layers (closer to the output) have larger learning rates. This strategy is based on the idea that the initial layers capture more generic features that are less likely to change significantly, whereas the later layers capture more task-specific features that require more significant adjustments. When the model has stronger emergence according to our theory, it is more likely to learn the specific tasks faster in a fine tuning process.

Now we aim at proposing a network initialization architecture with stronger emergence. To do so, we decrease the weight magnitude in the first half of the layers and increase the weight magnitude in the second half of the layers.

- Decrease the activity of nodes in the first half layers
- Increase the activity of nodes in the second half layers

in order to achieve this, we design the following initialization scheme:

- Decrease the magnitude of weights in the first half layers by dividing a factor $\alpha$

- Increase the magnitude of weights in the second half layers by multiplying a factor $\alpha$

because in general, a larger weight magnitude can lead to higher activation thus increase the number of active nodes.

We also want to consider the stability of the network. The existing initialization schemes were usually designed such that the activation and gradients are stable across the layers. For example, in Xavier initialization, it is shown that

$$n_i Var(W_i) = 1 \tag{8}$$
$$n_{i+1} Var(W_i) = 1 \tag{9}$$

promotes stability of activation and gradients. However, when we increase the variance in the inital layers and decrease the variance in the later layers, we introduce instability to the flow of activation and gradients across the layers. To reduce the effect of instability on the performance of initialization scheme, we make the increase of variance across the layers to be smooth, so as to reduce the instability. For example, we initialize the weight matrices in the following way:

We first initialize the network weights $\{W_i\}$ following some standard initialization scheme which preserves stability, for example, Xavier or Kaiming He Initialization. Then we do the following scaling to the weights:

$$
\begin{aligned}
\tilde{W}_{-n} &= W_n/\alpha^n \\
\tilde{W}_{-(n-1)} &= W_{-(n-1)}/\alpha^{n-1} \\
&\vdots \\
\tilde{W}_0 &= W_0 \\
\tilde{W}_1 &= W_1 * \alpha \\
&\vdots \\
\tilde{W}_{n-1} &= W_{n-1} * \alpha^{n-1} \\
\tilde{W}_n &= W_n * \alpha^n
\end{aligned} \tag{10}
$$

In our experiments, we see in Figure 3 that this initialization is indeed leading to a better performance. In particular, we show the correlation between emergence and performance. Based on our theory, we have an increase in emergence, even when only the magnitude of weights of the first half layers decreases, or the magnitude of weights of the second half increases.

We also note that, such choice of layer magnitudes is mimicking the "domino effect", as illustrated in the figure below. The increase of energy level for each piece can set off a cascade effect.

We now study how to choose the scaling factor $\alpha$ properly. As shown in our numerical experiments, we can see that as $\alpha$ increase, the performance first increases then decrease, and the decrease part is likely to be caused by the instability inherent to the initialization scheme. In order to determine a factor $\alpha$ that is appropriate, we want to limit the emergence of the model to a range.

**Choice of optimal $\alpha$:** From the mathematical equation of emergence, we can see that given a model, the maximum amount of emergence is determined by the parameters of the equations, $N$ and $n_i$, which are the number of layers in the network and the size of each layer. Emergence increases in $O(n_i)$ and $O(N^2)$. As a result, emergence is more sensitive to $\alpha$ when the network has more layers. So we allow larger $\alpha$ when the network is shallow and smaller $\alpha$ when the network is deep. Empirically, under the learning rate $lr = 0.001$, $\alpha = 2$ is a good choice for usual architectures. For the two layer MLP, for example in transformers, $\alpha$ can be as large as 10, which for deeper MLP, $n > 5$, then smaller $\alpha$ should be considered.

Here our motivation is simply to bound the emergence value. We should also bear in mind that stability is also a very important issue for an initialization scheme to behave well. We encourage people to give more rigorous analysis on emergence and stability so as to strike a more optimal balance between these two.

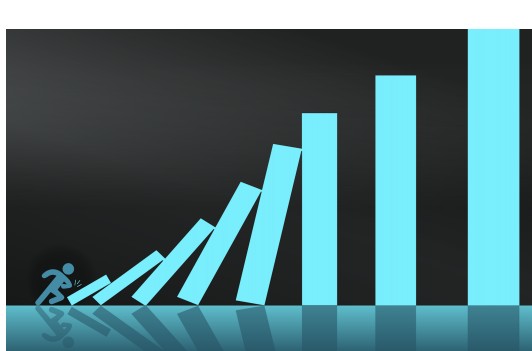 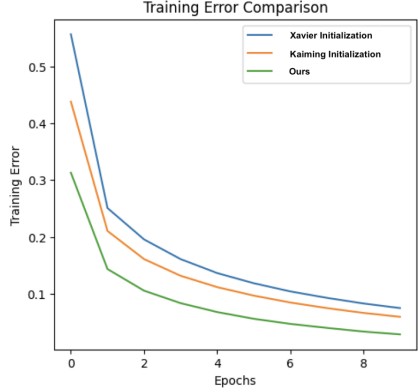

Figure 2: (A) The increase of weight magnitude through the layers mimicks how the energy level through the pieces of a domino is increasing. (B) Comparison of training loss and emergence of Xavier, Kaiming and our initialization schemes. Emergence measure: Xavier: $5.03e^8$, Kaiming: $5.99e^8$, Ours: $10.87e^8$

## 4 EXPERIMENTS

We evaluate our initialization scheme on benchmark datasets for image classification and machine translation tasks. For image classification, five different architectures are evaluated for CIFAR-10 (Krizhevsky et al., 2009), and ResNet-50 (He et al., 2016) is evaluated for ImageNet(Deng et al., 2009). For machine translation, we use our initialization scheme to find good initializations for a Post-LN Transformer without any change to its original architecture on IWSLT-14 De-En (Cettolo et al., 2014).

We conduct our experiments in PyTorch. We use the fairseq library for machine translation (Ott, 2019). All the experiments on CIFAR-10 and IWSLT-14 DE-EN can run with one single NVIDIA A100 GPU. Our initialization scheme first initializes the weights using Kaiming initialization for all the Conv and MLP layers for image classification. On ImageNet, we compare with Kaiming Initialization, and GratInit. Each block in ResNet-50 is initialized independently with $\alpha = 2$. We use batch normalization to increase stability.

For machine translation, we use the default Xavier initialization (Glorot & Bengio, 2010). Base on the discussion in the previous section, we choose the scale factors $\alpha = 2$ with out batch normalization and $\alpha_i = 5$ with batch normalization.

On CIFAR-10, we focus on MLP and the feedforward VGG net with and without BN layers. Since ResNet has recursive network structure, we leave it to a future work to establish the emergence formula on it. For MLP, we use a simple MLP architecture with 3 hidden layers. For VGG net, we use VGG-19 and our initialization scheme is compared with four different methods/settings: 1) Kaiming Initialization (He et al., 2015); 2) First train the network for one epoch with a constant learning rate equal to the starting learning rate, labelled as "+1 epoch (Const. LR)" in Table 1; 3) First train the network for one epoch with a linear warmup learning rate, labbeled as "+1 epoch (Warmup)" in Table 1; 4) MetaInit (Dauphin & Schoenholz, 2019). The data is from GradInit(Zhu et al., 2021). On CIFAR-10, we train networks with a batch size of 128, and in our initialization scheme, we adopt a constant learning rate of $0.001$, while in other initialization models, much larger learning rate (for example, $0.1$) has been adopted. Our scheme has significant performance even though the learning rate is much smaller.

From our experiments we can also see BN does stabilize VGG-19 and allows training with stronger emergence (larger value of $\alpha$). This shows the particular promising application of our scheme combined with batch normalzation. We can see from our numerical simulation that since batch normalization promotes good stability, we are free to choose larger $\alpha$.

IWSLT'14 DE-EN (Cettolo et al., 2014) is a German to English translation dataset that has 160k training examples. Our Transformer model is inherited from (Vaswani, 2017), which is a Post-LN

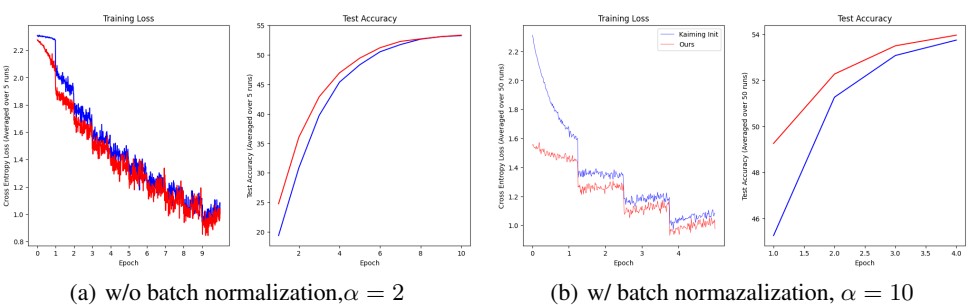

(a) w/o batch normalization, $\alpha = 2$         (b) w/ batch normazalization, $\alpha = 10$

Figure 3: Training loss and test accuracy of MLP on CIFAR-10.

Table 1: First epoch (Acc1) for models on CIFAR-10, $\alpha = 2$ for VGG-19 without BN, and $\alpha = 5$ for VGG-19 with BN.

| Model | VGG-19 w/o BN | VGG-19 w/ BN | ResNet-110 w/o BN | ResNet-110 w/ BN | ResNet-1202 w/ BN |
|---|---|---|---|---|---|
| Kaiming | $29.1 \pm 1.5$ | $12.6 \pm 0.6$ | $16.1 \pm 2.1$ | $23.2 \pm 0.9$ | $12.9 \pm 2.8$ |
| +1 epoch (Const. LR) | $37.2 \pm 1.1$ | $19.6 \pm 4.0$ | $21.0 \pm 3.8$ | $32.5 \pm 3.8$ | $12.6 \pm 2.8$ |
| +1 epoch (Warmup) | $37.4 \pm 1.2$ | $53.5 \pm 2.9$ | $19.8 \pm 0.5$ | $48.7 \pm 1.1$ | $28.1 \pm 1.3$ |
| MetaInit | $30.5 \pm 0.9$ | $35.1 \pm 0.6$ | $14.6 \pm 2.2$ | $29.0 \pm 1.5$ | $11.7 \pm 1.6$ |
| GradInit | $29.3 \pm 0.6$ | $47.8 \pm 1.8$ | $36.2 \pm 0.8$ | $38.2 \pm 0.9$ | $29.0 \pm 1.1$ |
| Ours | $\mathbf{46.2} \pm 0.6$ | $52.4 \pm 1.0$ | $\mathbf{45.3} \pm 2.0$ | $48.0 \pm 1.5$ | $\mathbf{29.8} \pm 1.7$ |

Table 2: Accuracy after epoch 1 of ResNet-50 models on ImageNet. Results from (Zhu et al., 2021).

| Model | Kaiming | GradInit | Ours |
|---|---|---|---|
| $Acc_1$ | 14.6 | 19.2 | 23.2 |

Table 3: A comparison of Emergence-Promoting Initialization with other initialization for training the Post-LN Transformer model on the IWSLT-14 De-EN dataset. (Evaluate after 80 epochs)

| Model | $BLEU_1$ | $BLEU_{best}$ |
|---|---|---|
| Xavier | - | 34.85 |
| T-Fixup | 3.96 | 34.78 |
| Ours | 4.8 | 35.13 |

Transformer placing its Layer Normalization after the summation of the skip connection and the residual branch. It has a $512$- dimensional word embedding layer and 1024 dimensions in its hidden FFN layer. It has 6 encoder layers and 6 decoder layers. We choose the learning rate to be $5e-4$ with inverse-sqrt learning schedule with $4000$ warmup updates and weight decay of 0.0001.

Based on the discussion in the previous section, we have two ways of promoting emergence: A) promote emergence globally, by reducing the magnitude of weights in the encoder layers and increasing the magnitude of weights in the decoder layers; B) promote emergence clockwise, where we apply (8) to each encoder/ decoder block. In our transformer architecture, there is MLP block in each encoder/decoder layer consisting of 2 layers. In our experiments, we can choose $\alpha$ up to 10 for each MLP block, and notably we can see fast increase of BLEU score in the first few epochs. For example, with $\alpha = 10$, the BLEU score after first epoch reaches 6.02 while for T-Fixup the BLEU score after first epoch is 3.79.

For global emergence promoting scheme, we first initialize our scheme based on T-Fixup, and then increase the magnitude of weights in the decoder layers by 2. We run the models for the maximum of 80 epochs and evaluate the BLEU score every epoch, and report the best BLEU scores throughout training for each run and the result is in Table 3.

In all our numerical experiments, we notice our initialization leads to better training performance when combined with batch normalization, weight decay and other techniques that promotes stability and prevents over-fitting, while in other cases the model could be trapped in local minimum. We encourage researchers to combine our initialization scheme with other stability-promoting considerations, which could potentially further improve the performance (especially long term) of our initialization.

## 5 CONCLUSION

In this paper, we introduced a novel and straightforward neural network initialization scheme inspired by the concept of emergence. Building on the emergent network measures proposed by (Li et al., 2023), our method adjusts the layer-wise variance parameters to enhance the number of paths from inactive to active nodes, thereby achieving higher emergence values. This approach is not only easy to implement but also requires no additional optimization or training steps compared to conventional methods like Xavier and Kaiming initialization. Our extensive evaluations across various architectures, including convolutional neural networks (CNNs) for image recognition and transformers for machine translation, demonstrate the significant advantages of our initialization scheme. The empirical results show that our method substantially improves model accuracy and convergence speed on standard datasets such as CIFAR-10, ImageNet and the IWSLT-14 translation task.

Our work contributes to the growing body of research that seeks to understand and harness the power of emergence in neural networks. By providing a simple yet powerful modification to existing initialization techniques, we open new avenues for improving neural network training methodologies. Future work could explore further optimizations and adaptations of our initialization scheme to other types of neural architectures and more complex tasks. Overall, emergence-promoting initialization scheme represents an addition to current neural network initialization practices, offering both theoretical insights and practical improvements for the development of more robust and capable machine learning models.

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

# A APPENDIX

## A.1 MATHEMATICAL REPRESENTATION OF NEURAL NETWORKS

To perform mathematical computation of emergence, we use quiver representation as the representation of a neural network. Formally, a **quiver** is a directed graph where loops and multiple arrows between two vertices are allowed, defined as follows:

- A quiver $Q$ is a quadruple $Q = (Q_0, Q_1, h, t)$ where $Q_0$ is a finite set of vertices, $Q_1$ is a finite set of arrows, and $h$ and $t$ are functions $Q_1 \rightarrow Q_0$. For an arrow $a \in Q_1$, $h(a)$ and $t(a)$ are called the head and tail of $a$.
- We get a **representation** $V$ of $Q = (Q_0, Q_1, h, t)$ if we attach to every vertex $x \in Q_0$ a finite dimentional vector space $V(x)$ and to every arrow $a \in Q_1$ a linear map $V(a) : V(ta) \rightarrow V(ha)$.

Quiver representation can be used to model the dynamics on the network Derksen & Weyman (2017); Armenta & Jodoin (2021); Armenta et al. (2023). We provide two examples of quiver representation in Figure Figure A.1.

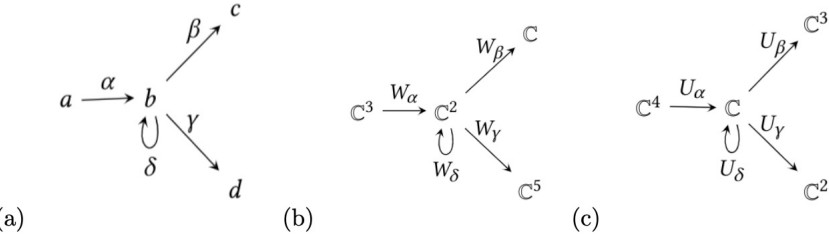

(a)  (b)  (c)

Figure 4: Additional examples of quivers. (a): A quiver $Q$ with vertices $V = \{a, b, c, d\}$ and oriented edges $E = \{\alpha, \beta, \gamma, \delta\}$, (b) and (c): two quiver representations over $Q$. Adapted from (Derksen & Weyman, 2017).

**Theorem** (Proposition 5.3 in (Li et al., 2023)) Given the functor $\Phi$ which preserves partial structure in a quiver representation $W$ by deleting a set of edges $E$, the derived functor of $\Phi$ is

$$R^1\Phi(W) = \bigoplus_{a\in E} \Phi(W(ta) \otimes P_{ha}) \tag{11}$$

where $ta$ is the tail of edge $a$ (the starting node), $ha$ is the head of edge $a$ (the ending node), $W(ta)$ is the vector space associated to node $ta$, $P_{ha}$ is the vector space spanned by all paths originating from node $ha$.

*Proof.* Based on (Derksen & Weyman, 2017), for representation $W$ in **Rep**$(Q)$ we have the projective resolution

$$0 \longleftarrow W \xleftarrow{\ f^W\ } \bigoplus_{x\in Q_0} W(x) \otimes P_x \xleftarrow{\ d^W\ } \bigoplus_{a\in Q_1} W(ta) \otimes P_{ha} \longleftarrow 0 \tag{12}$$

where

$$f^W : \bigoplus_{x\in Q_0} W(x) \otimes P_x \to W \tag{13}$$

is defined by

$$f^W(w \otimes p) = p \cdot w, \tag{14}$$

and

$$d^W : \bigoplus_{a\in Q_1} W(ta) \otimes P_{ha} \to \bigoplus_{x\in Q_0} W(x) \otimes P_x \tag{15}$$

is defined by

$$d^W(w \otimes p) = (a \cdot w) \otimes p - w \otimes pa. \tag{16}$$

Now we compute the first left derived functor $R^1\Phi$. By definition (Rotman & Rotman, 2009), it is the 1st homology object of the sequence above under the image $\Phi$, formally, $R^1\Phi = \ker \Phi d^W$, where $d^W$ is defined in (15). Now if an edge $a$ is deleted by the functor $\Phi$ then for any $w \in W(ta)$ and $p \in P_{ha}$, we have $(a \cdot w) \otimes p = w \otimes ap = 0$, hence $\Phi(W(ta) \otimes P_{ha}) \subseteq \ker \Phi d^W$. If $a$ is preserved under $\Phi$, then $\Phi d^W$ will act the same as $d^W$ on $\Phi(W(ta) \otimes P_{ha})$, and $d^W$ is injective due to the exactness of resolution, $\Phi(W(ta) \otimes P_{ha})$ will be non-zero thus not contribute to $\ker \Phi d^W$.

$\square$

This theorem computes $R^1\Phi$, which evaluates the difference between $\Phi(s_1 \vee s_2)$ and $\Phi(s_1) \vee \Phi(s_2)$, thus encodes the potential of a system for emergence. Taking advantage of this theorem, we can take the dimension of $R^1\Phi(W)$ as a numerical approximation of the potential for emergence of $W$ when the network interact with other networks:

$$\dim R^1\Phi_l(W) = \dim \bigoplus_{e\in E} \Phi_l(W(he) \otimes I_{te})$$

$$= \sum_{e\in E} \dim \Phi_l(W(he)) \times \dim \Phi_l(I_{te}). \tag{17}$$

Here $\dim \Phi_r(V(te))$ and $\dim \Phi_l(W(he))$ is the dimension of the image of the vector space $V(te)$ and $W(he)$ under the functor, and $\dim \Phi_r(P_{he})$ and $\dim \Phi_l(I_{te})$ is the dimension of the image of the path algebra $P_{he}$ and $I_{te}$ under the functor.

Given a network $G$, and a sub-network $H$ which represents its effect or observation under $\Phi$, where their relation are shown as follows:

$$G \xrightarrow{\text{Cross-scale mapping } \Phi} H$$

then we have the following measure of emergence for networks:

$$\text{Emergence}(G, H) = \sum_{x \in G \setminus H} \#\text{paths in } H \text{ from } N_H(x) \text{ to } H, \tag{18}$$

where $H$ represents the part of network structure being preserved by $\Phi$, the partial observation. $N_H(x)$ is the set of downstream neighbors of $x$ in $H$.

## A.2 HYPERPARAMETERS

**On learning rates** Empirically, the scaling factor $\alpha$ is also dependent on learning rate. When the learning rate is faster, more stability is usually required and hence we need smaller emergence. There has been some theoretical results on the optimal learning rate, for example, (Hettinger, 2019) suggested that the optimal learning rate should be inversely proportional to the gradient magnitude at initialization:

$$\eta = \frac{c}{\|\nabla L\|} \tag{19}$$

where $\eta$ is the optimal learning rate, $\|\nabla L\|$ is the magnitude of the gradient of the loss function $L$ with respect to the network parameters at initialization, and $c$ is a constant.

So when given the learning rate, we should choose $\alpha$ such that the resulting gradient magnitude at initialization is inversely proportional.

If we assume that by introducing our scheme we have $C(\alpha)^N$ times increase to the initial gradient.Then under the learning rate $\eta$ we have

$$\frac{\eta}{\eta_0} = \frac{\|\nabla L_{\alpha_0}\|}{\|\nabla L_\alpha\|} = \frac{\alpha_0^N}{\alpha^N} \tag{20}$$

hence we have

$$\alpha = \alpha_0 \left[\frac{\eta_0}{\eta}\right]^{1/N}. \tag{21}$$

For a two layer network, if we choose $\alpha_0 = 2$ for learning rate $\eta_0 = 0.001$, then for a different learning rate $\eta = 0.0001$ we should choose $\alpha = 6.32$.

Since based on our scheme, the initial gradient varies across layers, the layer-wise learning rates configuration should also be considered a good choice. Specifically, the learning rate for each layer, denoted by $\eta_l$, should be proportional to the inverse of the square root of the expected squared gradient norm at initialization. Mathematically, this can be expressed as:

$$\eta_l \propto \frac{1}{\sqrt{E\left[\|\nabla L(\mathbf{x}_l^0)\|^2\right]}}$$

where $\eta_l$ is the learning rate for layer $l$, and $E\left[\|\nabla L(\mathbf{x}_l^0)\|^2\right]$ is the expected squared gradient norm of the loss $L$ with respect to that layer's inputs $\mathbf{x}_l^0$ at initialization. This approach aims to optimize the learning process by adjusting the learning rates according to the variability and scale of the gradients encountered in different layers of the neural network.

**Other architectures and block wise initialization** Note that (9) only works for MLP, which has good symmetry. For convolutional layers, we can modify (9) to get the following measure of emergence:

$$E = \sum_{\substack{i=1}}^{N-1} \sum_{\substack{j>i}}^{N} (n_i - a_i)a_j \prod_{k=i+1}^{j-1} m_k \tag{22}$$

where $m_k$ is the number of filters in layer $K$. The analysis largely follows.

In most convolutional architectures and transformers, there are MLP blocks presented. While we can only do our internalization to the MLP blocks and see an improvement, we can also consider applying scheme (19) to the convolutional layers. Given this formula for emergence, we can increase the global emergence, but also increase the local emergence by applying scheme (19) to some of the layer blocks. We will discuss this in the next section.

