# OpenReview forum: "Advancing Neural Network Performance through Emergence-Promoting Initialization Scheme"
_ICLR.cc/2025/Conference — Submitted to ICLR 2025_

### Official Review · Reviewer_PoKs · 2024-10-30

**Soundness:** 2
**Presentation:** 2
**Contribution:** 2
**Rating:** 3
**Confidence:** 4

**Summary:**

This paper explores emergence property of neural network and introduce a NN initialization scheme that aims at achieving better potential for emergence. The authors conduct experiments on MLP and convolutional architectures for image recognition, and transformers for machine translation to evaluate the proposed approach.

**Strengths:**

1. Searching for a network initialization method that can promote the emergence phenomenon is meaningful, which helps to accelerate model convergence and improve generalization performance.

2. The proposed method achieves better performance in initial training phase across several tasks.

**Weaknesses:**

1.The experimental results can not support the claims made in the paper. As we know, In the field of machine learning, particularly in deep learning and large neural networks, the phenomenon of emergence describes how models exhibit abilities or characteristics they did not originally possess or that were not anticipated, after scaling up in size, increasing data volume, or extending training time.
However, from the experimental results presented, the only information I can get is that the method proposed in this paper converges faster in small models or datasets during the initial phase (the first epoch) of training compared to Xavier or Kaiming initialization. However, it does not demonstrate stronger emergent properties or scaling capabilities. I suggest that the authors train on larger datasets and models for longer durations to support the claims made in the paper.

2.There are many metrics and experiments that can measure emergent phenomena, such as capturing performance jumps in zero-shot and few-shot learning and conducting scaling laws analysis. However, this paper only presents training loss and test accuracy. I suggest that the authors provide a more comprehensive evaluation of their proposed method.


minor typo:
L274: ‘nods’ should be ‘nodes’.

**Questions:**

What is the meaning of operations $\bigoplus$ and $\otimes$？ I think this paper should illustrate them.

---

### Official Review · Reviewer_KAq8 · 2024-11-02

**Soundness:** 2
**Presentation:** 2
**Contribution:** 2
**Rating:** 3
**Confidence:** 4

**Summary:**

The paper introduces a straightforward neural network initialisation scheme to enhance the network's potential for emergent behaviours. Specifically, it adjusts the weight initialisation by reducing the weight magnitudes in the first half of the layers while increasing them in the second half.

**Strengths:**

1- Investigating the emergent behaviour of neural networks, particularly in deep models, opens new avenues for advancement and shows promising potential.

2- The proposed method seems straightforward and convenient to implement.

**Weaknesses:**

1-The paper can be improved by improving the writing and presentation of the work. Some figures, like Fig 1 and Fig 2(a), are less informative regarding technical and professionalism. The figures' quality should be improved. They are too small and hard to read.

2-The evaluation and analysis provided are limited. A more in-depth ablation study on the choice of the turning point for decreasing and increasing weight magnitudes is essential.

3-The primary evaluation focuses on image classification during the first epoch of training. However, a good initialisation is also crucial for guiding the model toward improved final performance.

4- How the proposed method affects the convergence speed is unclear.

5 How can changing the learning rate influence the proposed initialisation schema?

**Questions:**

Please refer to weaknesses.

---

### Official Review · Reviewer_bKVQ · 2024-11-05

**Soundness:** 2
**Presentation:** 1
**Contribution:** 1
**Rating:** 3
**Confidence:** 3

**Summary:**

From the perspective of emergence-promoting, the paper presents a method to initialize the neural networks for training. Starting from abstract definition of emergence, the authors view a neural network as a graph, and derive the definition of emergence in a neural network based on the number of paths from inactive nodes in lower layers to active nodes in higher layers. It turns out that the number of inactive nodels determine the emergence of a neural network. Then the authors propose a simple method to initialize the neural network by decreasing the magnitude of weights in the first half layers and increasing the magnitude of weights in the second half layers. Experiments were conducted on CIFAR-10, ImageNet and IWSLT-14 De-En.

**Strengths:**

The perspective of emergence for neural network initialization is interesting. The derived result, say, decreasing the magnitude of weights in the first half layers and increasing the magnitude of weights in the second half layers, is easy to implement. Experiments showed that the neural networks initialized with this strategy had some advantage.

**Weaknesses:**

1. The transition from the Theorem (L200) to equation (3) is hard to understand. In other words, how eqn (3) which describing a concrete example, is derived from eqn (2) which describes an abstract system, is unclear. More explanation is needed.

2. The contents after eqn (3), say, L231-241, are inconsistent with this eqn. In this eqn, H belongs to G, but in L231-241, G and H denote two different sets of nodes. So, I cannot understand these contents.

3. How eqn (4) is derived from eqn (3) is also unclear. The authors are suggested to make it clear what G, H, N_H(x) are in eqn (4).

4. Experiments only showed that during the initial training epochs, e.g., the first epoch in Table 1 and Table 2,  the proposed method performed well. But in most applications, people want the final training result to be good. Table 3 shows the result after 80 epochs but the result is only slightly better than Xavier (35.13 versus 34.85).

5. The presentation of the paper is poor. Section 4 is not written in a logical manner. The figures and tables are not cited in appropriate places. Readers even don't know what results correspond to these figures and tables. Table 2 is not cited in the main text. In L344-355, what are the subscripts -n and -(n-1)? In eqn (2) what are the special symbols of plus and multiplication?

**Questions:**

All of above weakness points are important for improving the paper.

---

### Meta-Review · Area_Chair_H2Vs · 2024-12-23

**Metareview:**

This paper proposes a neural network initialization scheme by adjusting the layer-wise weight scaling factors to improve the conventional methods Xavier and Kaiming initialization. Experiments on various architectures are conducted. However, multiple reviewers raise concerns regarding the poor presentation and the unconvincing experimental results. The authors did not provide response in the rebuttal so the concerns are not resolved. The AC recommends reject.

**Additional Comments On Reviewer Discussion:**

no response from the authors, and no discussion during the rebuttal period.

---

### Decision · Program_Chairs · 2025-01-22

Reject